# Sustained Maternal Smoking Triggers Endothelial-Mediated Oxidative Stress in the Umbilical Cord Vessels, Resulting in Vascular Dysfunction

**DOI:** 10.3390/antiox10040583

**Published:** 2021-04-09

**Authors:** Szabolcs Zahorán, Péter R. Szántó, Nikolett Bódi, Mária Bagyánszki, József Maléth, Péter Hegyi, Tamás Sári, Edit Hermesz

**Affiliations:** 1Department of Biochemistry and Molecular Biology, Faculty of Science and Informatics, University of Szeged, H-6701 Szeged, Hungary; zahoran.szabolcs@gmail.com (S.Z.); szanpet96hu@gmail.com (P.R.S.); 2Department of Physiology, Anatomy, and Neuroscience, Faculty of Science and Informatics, University of Szeged, H-6701 Szeged, Hungary; bodi.nikolett@bio.u-szeged.hu (N.B.); bmarcsi@bio.u-szeged.hu (M.B.); 3First Department of Medicine, University of Szeged, H-6701 Szeged, Hungary; maleth.jozsef@med.u-szeged.hu; 4HAS-USZ Momentum Epithel Cell Signalling and Secretion Research Group, H-6701 Szeged, Hungary; 5Institute for Translational Medicine, Medical School, University of Pécs, H-7601 Pécs, Hungary; hegyi.peter@med.u-szeged.hu; 6Department of Obstetrics and Gynaecology, Faculty of Medicine, University of Szeged, H-6701 Szeged, Hungary; drsari.szeged@gmail.com

**Keywords:** arginase, endothelial dysfunction, maternal smoking, metal exposure, nitric oxide synthase, xanthine oxidoreductase

## Abstract

Nitric oxide (NO) bioavailability is fundamental in the regulation of redox balance and functionality of the endothelium, especially in the case of the umbilical cord (UC), which has no innervation. The analysis of UC vessel-related complications could serve as a useful tool in the understanding of the pathophysiological mechanisms leading to neonatal cardiovascular disorders. Therefore, the aim of this study was to characterize the mechanisms that rule the severity of prenatal endothelial dysfunction, induced by the long-term effect of maternal smoking. Our analysis describes the initiation and the consequences of endothelial nitric oxide synthase (NOS3) deactivation, along with the up-regulation of possible compensatory pathways, using structural, molecular and biochemical approaches. This study was carried out on both the UC arteries and veins originated from neonates born to non-smoking and heavy-smoking mothers. The alterations stimulated by maternal smoking are vessel-specific and proportional to the level of exposure to harmful materials passed through the placenta. Typically, in the primarily exposed veins, an increased formation of reactive oxygen species and an up-regulation of the highly-efficient NOS2-NO producing pathway were detected. Despite all the extensive structural and functional damages, the ex vivo heat and cadmium ion-treated UC vein pieces still support the potential for stress response.

## 1. Introduction

The endothelium plays a complex role in the vascular system. It contributes to the vascular homeostasis, and its functional and morphological alterations take part in the formation of pathophysiological processes. The effects of the traditional cardiovascular risk factors, like diabetes, smoking, and dyslipidemia, on the adult cardiovascular system have been well-known and are associated with endothelial dysfunction (ED). Though less is known about the molecular consequences/background of fetal exposure to toxic materials, there is increasing evidence that environmental agents can adversely affect the in utero development and even mediate long-lasting health consequences. A chemo-biology interactome analysis underlined this concept and provided evidence that the components of tobacco smoke intensively influence gene expression at both the embryonic and fetal developmental periods [1,2,3]. During pregnancy, the placenta and the umbilical cord (UC) form vital temporary organs to sustain a proper fetal development. Therefore, any alteration in the in utero condition is directly or secondarily associated with the placental or UC disorders, like intrauterine hypoxia and an impaired blood flow to the fetus [4,5,6]. The UC vein carries oxygenated blood and nourishment from the placenta to the fetus, and the two UC arteries transport deoxygenated blood back to the placenta. Therefore, UC vessels are directly and primarily exposed to harmful substances passing through the placenta. The UC blood vessels lack innervation, so the tone of the vessels is mainly governed by local regulators in active communication with the milieu (endocrine factors, shear stress, oxygen concentration, etc.) [7,8,9,10]. Nitric oxide (NO), the highly diffusible vasodilator molecule, is synthetized by nitric oxide synthase family members. In endothelial cells, the main isoform is the endothelial nitric oxide synthase (NOS3), whose activity is well coordinated and can be compromised at many points. NOS3 is localized in the cytoplasm and the caveolae in the plasma membrane, and its activation is regulated by dimerization, posttranslational modifications and protein-protein interactions [11]. One of the main starting points for ED is the reduced production/bioavailability of NO, resulting in increased oxidative stress in the endothelium. The oxidative milieu induces NOS3 homodimer uncoupling, resulting in the formation of superoxide anion (O_2_^•−^) instead of NO [12]. The elevated O_2_^•−^ level also triggers the upregulation of arginase. Arginase-1 (ARG1) in competition distracts the common L-arginine substrate, which indirectly leads to further NOS3 uncoupling and reactive oxygen species (ROS) formation [13,14]. Another major regulatory point in NOS3 activation is phosphorylation/dephosphorylation at several amino acid side chains. It is generally accepted that phosphorylation at the serine 1177 side chain (P-NOS3) is a crucial requirement for activation [15]. In the case of reduced NOS3-dependent NO production / bioavailability, alternative NO producing pathways get activated as compensatory mechanisms to improve the blood flow to the fetus. The vessel endothelium is in continuous contact with circulating red blood cells (RBCs), and, moreover, an intimate crosstalk between them was recently reported under pathological conditions [16,17]. Supposing that ED can be sensed by circulating RBCs, NO bioavailability might be increased by the RBC-NOS3 activation pathway. However, our previous studies on the maternal and fetal RBCs indicated significant alterations in the RBCs’ morphological parameters, elastic and plastic properties and abnormalities in their membrane-lipid composition, along with an impaired NOS3 activation, as the consequence of maternal smoking. All these alterations make it most likely that the NOS3-dependent NO production by RBCs is not available as a compensatory mechanism [16,18,19]. Another obvious solution for compensation could be the upregulation of inducible nitric oxide synthase (NOS2) expression in the endothelial layer, due to its high (100–1000 fold that of NOS3) catalytic activity and/or the activation of xanthine oxidoreductase (XOR) [20,21]. Though this second assumption might sound astonishing, knowing the basic activity of XOR, recent studies presented evidence on the beneficial outcomes of XOR activation by the reduction of biologically inert nitrite (NO_2_)/nitrate (NO_3_) to NO [22,23,24]. Our datasets on the artery and vein endothelium clearly demonstrate that the maternal-smoking-induced alterations are vessel-specific and most likely a close reflection of the in vivo circumstances of prenatal development. Furthermore, we provide evidence to show that in a highly hypoxic environment, with a low bioavailability of NO, the elevated NO_2_/NO_3_ pool unravels the recently described protective role of XOR.

## 2. Materials and Methods

### 2.1. Sample Collection

UC samples were collected at the Department of Obstetrics and Gynecology at the University of Szeged, Hungary. Samples from informed volunteers were handled according to the Declaration of Helsinki, and The Ethics Committee of the Department of Obstetrics and Gynecology approved our study protocol (16/2016). UC fragments from neonates born to heavy-smoking mothers (at least 10 cigarettes per day during the entire pregnancy period) (nSm = 12), along with samples from age-matched neonates born to non-smoking mothers, were analyzed (nCtrl = 12). According to our study protocol for sample collection, strict exclusion factors were selected, such as maternal age below 18 years, gestational age below 37 weeks, infection and inflammatory conditions, gestational diabetes, high blood pressure, stroke, atherosclerosis, heart failure, drug treatment, intrauterine distress, malformations or evidence of genetic disorders. UCs were transported to the laboratory immediately after delivery and processed for further use.

### 2.2. Tissue Preparation and Transmission Electron Microscopy

The dissected umbilical segments were processed for transmission electron microscopy. Briefly, small pieces of the umbilical segments were fixed overnight at 4 °C in a 2% paraformaldehyde and 2% glutaraldehyde solution, and then further fixed for 1 h in 1% osmium tetroxide. After rinsing in buffer and dehydrating, they were embedded in Embed812 (Electron Microscopy Sciences, Hatfield, PA, USA). The blocks were used to prepare semithin (0.7 µm) sections to select the area of interest, then ultrathin (70 nm) sections, which were mounted on a nickel grid. The sections were counterstained with uranyl acetate and lead citrate, and then examined and photographed with a JEOL 1400 transmission electron microscope (JEOL Ltd, Tokyo, Japan).

### 2.3. Viability and Apoptosis Detection Assay

The evaluation of viability was carried out on freshly isolated human umbilical vein endothelial cells (HUVEC) with the Annexin V-FITC Apoptosis Staining/Detection Kit: ab14085, Abcam (Cambridge, UK), according to the manufacturer’s protocol. In short, HUVECs were mechanically isolated after introducing a specially designed cannula into the umbilical veins. The cells were washed with a 0.9% (*w*/*v*) salt solution, and the suspension was centrifuged (4 °C, 2400 g, 10 min). Cell pellets were re-suspended in a 200 µL 1 × Binding Buffer, and 2 µL of Annexin V-FITC and 5 μL of propidium iodide were added to the suspension. After 5 min of incubation at room temperature, the samples were processed with quantitative flow cytometry (FACS) (BD FACSCalibur™, BD Biosciences). The results were analyzed using FlowJo™ (FlowJo™ Software for Windows [software application] Version 10. Ashland, OR, USA: Becton, Dickinson and Company; 2019). The cells were gated out according to their side and forward scatter properties, and then the fluorescent threshold was specified to our non-labelled HUVEC suspension. Apoptotic and healthy cells were distinguished based on the FL-1 (FITC) versus FL-3 (propidium iodide) quadrants within each population. The purity of the isolated HUVEC population was verified with monoclonal mouse anti-von Willebrand factor (vWF) (F8/86): sc-53466 from Santa Cruz Biotechnology (Dallas, TX, USA) and polyclonal rabbit anti-Phospho-NOS3 (pSER1177): SAB4300128 Sigma-Aldrich (St. Louis, MO, USA) double immunolabeling, followed by FACS analysis [25,26].

### 2.4. Ex Vivo Reactivity Test on Isolated UC Vessels

UC veins from the non-smoking and smoking groups were dissected on ice, within 1 h after delivery, and split into 23 cm pieces. Vein pieces used as absolute controls (Abs 0), immediately frozen in liquid nitrogen and stored at −80 °C. To model the acute heavy metal exposition, the vein fragments were transferred into 12-well cell culture plates containing Dulbecco’s Modified Eagle Medium (with 4.5 g/L glucose and with l-glutamine, DMEM, Lonza Bioscience, Basel, Switzerland: 12-604F) with and without 0.5 ng/µL and 2.5 ng/µL Cd^2+^ in the form of cadmium acetate dihydrate (Sigma – Aldrich, St. Louis, MO, USA: 289159) in a humidified incubator at 37 °C and 5% CO_2_. Cd^2+^ exposure was carried out for 30 min, and then the vein fragments were rinsed twice with DMEM and further incubated for 30 min in a Cd^2+^-free DMEM solution. The fragments incubated in the Cd^2+^-free DMEM solution for 1 h were considered as heat shock controls. At the end of the incubations, the vein fragments were frozen in liquid nitrogen and kept at −80°C until RNA extraction [27].

### 2.5. RNA Extraction, Reverse Transcription and Real-Time Quantitative PCR (qPCR)

Frozen vessels were pulverized under liquid nitrogen, homogenized in TRI-reagent (Zymo Research, Irvine, CA, USA: R2050-1-200), and total RNAs were prepared according to the Direct-zol RNA kit manufacturer’s recommendations (Zymo Research, Irvine, CA, USA: R2053), including genomic DNA elimination. RNA purity and concentration were measured with NanoDrop ND-1000 Spectrophotometer (Thermo Scientific, Waltham, MA, USA). First-strand cDNAs were synthesized using a 500 ng template and the Maxima H Minus First Strand cDNA Synthesis Kit with oligo(dT)18 priming, according to the manufacturer’s protocol (Thermo Scientific, Waltham, MA, USA: K1652). qPCR was performed on an Applied Biosystems 7500 Fast Real-Time system using Luminaris Color HiGreen qPCR Master Mix, Low ROX (Thermo Scientific, Waltham, MA, USA: K0371). The qPCR reactions were carried out in 96-well plates containing a 15 µL reaction mix/well with a temperature program of 95 °C for 10 min, then 40 cycles of 15 s at 95 °C and 60 s at 60 °C. The dissociation stage for one cycle (15 s at 95 °C, then 60 s at 60 °C and 15 s at 95 °C, and finally 15 s at 60 °C) was applied as an indication about the primer–target specificity. The Ct values of all samples were normalized to the internal control (18S rRNA), and the changes in mRNA levels were calculated by the ΔΔCt method [28]. The primers (Bio Basic Canada Inc., Markham, ON, Canada) were the following: 18S rRNA:forward (5’–3’): GAAACGGCTACCACATCCAAGGreverse (5’–3’): CCGCTCCCAAGATCCAACTACGheat shock protein 90 (hsp90):forward (5’–3’): CCGTTTCTGAGAAGCAGGGCAreverse (5’–3’): CCTTGGCTCTGTCTGAAGGC

### 2.6. Immunolabeling and In Situ Detection of Superoxide Anion on UC Cryosections

Small pieces of UCs were fixed in 4% (*w*/*v*) paraformaldehyde in a 0.05M phosphate-buffer (PB), cryoprotected with 30% (*w*/*v*) sucrose in PB supplemented with 0.1% (*w*/*v*) Na-azide. The specimens were embedded in Tissue-Tek^®^ O.C.T.™ (catalog number: 4583) obtained from Sakura Europe (Alphen aan den Rijn, Netherlands), cryo-sectioned (16 µm) and mounted on Superfrost™ ultra plus^®^ (J3800AMNZ) microscope slides from Thermo Scientific (Waltham, MA, USA). In the process of immunolabeling, the sections were permeabilized with 0.1% (*v*/*v*) Triton X-100 in PB, and non-specific antibody binding sites were blocked with 4% (*w*/*v*) bovine serum albumin (BSA) and 5% (*v*/*v*) normal goat serum (NGS) in PB. The sections were immunolabeled with primary monoclonal antibodies: mouse anti-NOS3 (A-9): sc-376751, mouse anti-Arginase-1 (C-2): sc-166920, mouse anti-inducible nitric oxide synthase (NOS2) (C-11): sc-7271, mouse anti-xanthine oxidase (A-3): sc398548 from Santa Cruz Biotechnology (Dallas, TX, USA), monoclonal mouse anti-4-hydroxy-2-nonenal [HNE]: ab48506, mouse anti-3-Nitrotyrosine (NT) (39B6): ab61392 from Abcam (Cambridge, UK) and polyclonal rabbit anti-Phospho-NOS3 (pSER1177): SAB4300128 Sigma-Aldrich (St. Louis, MO, USA). Primary antibodies were diluted uniformly in a ratio of 1:100, and the sections were incubated overnight at 4 °C. Secondary antibodies: goat anti-mouse or anti-rabbit IgG H & L conjugated with Alexa Fluor^®^ 488 or 647; ab150077; ab150113; ab150115; ab150079, respectively, from Abcam (Cambridge, UK). Secondary antibodies were diluted in a ratio of 1:1000, and the sections were incubated for 2 h at room temperature. All the dilutions were made with 1% (*w*/*v*) BSA and 5% (*v*/*v*) NGS in PB. Cell nuclei were counterstained with 1µg/mL 4’, 6-diamidino-2-phenylindole (DAPI) (D9542) from Sigma-Aldrich (St. Louis, MO, USA). The sections were mounted in an Antifading BrightMount/Plus aqueous mounting medium: ab103748, from Abcam (Cambridge, UK).

The O_2_^•−^ level was followed by in situ dihydroethidium staining (DHE) (D23107) Invitrogen™ (Carlsbad, CA, USA). DHE is a cell-permeable redox indicator, which can be oxidized in cytosol mainly by O_2_^•−^ anions and other strong oxidants, and then intercalates into a cell’s DNA and exhibits bright red fluorescence. In brief, DHE was applied at a 10 µM concentration, and the sections were incubated at 37 °C in a dark moist chamber for 30 min, washed with PB and then cover-slipped. The DHE’s specificity for O_2_^•−^ was tested with diethyldithiocarbamate (DETCA, catalog number: D3506) from the Sigma–Aldrich (St. Louis, MO, USA) treatment during the pilot experiments. DETCA, as a copper chelator, inhibits the internal superoxide dismutase activity, thus increasing the O_2_^•−^ concentration. In brief, consecutive UC sections were pre-incubated with and without DETCA (3 mM) in PB for 1 h at 37 °C in a dark moist chamber and washed three times, and DHE was applied as described above [29].

### 2.7. Imaging and Image Analysis

The sections were examined with a ZEISS LSM 880 confocal laser scanning microscope, equipped with Axiocam 503 mono (Carl Zeiss Microscopy GmbH, Germany). The image capturing was made with Zen 2.1 (black) software. During imaging, the arteries and veins of each sample were captured in at least 3–5 independent fields of view. For antibody-labeled sections, the following main parameters were applied: 8-bit images, Plan-Apochromat 40 × /1.4 Oil DIC M27 objective; excitation with Argon (488 nm) laser line for Alexa 488 fluorophore, Helium-Neon (633 nm) laser line for Alexa 647 fluorophore, violet (405 nm) laser line for DAPI; detection filters: 493 nm–591 nm, 630 nm–755 nm and 410 nm–479 nm; pinhole = 1 airy unit. For oxidized DHE detection, the following main parameters were applied: 8-bit images; alpha Plan-Apochromat 63 × /1.46 Oil Korr M27 objective; excitation with Argon (488 nm) laser line; detection filters: 517 nm–600 nm; pinhole = 1 airy unit. 8-bit pictures were semi-quantified with the ImageJ 1.50i software (NIH, USA) [30]. In brief, the composite 8-bit pictures were split into channels, corresponding to one specific antibody labeling. The layer of the endothelial regions was marked out by the freehand selection tool. The evaluation of the relevant fluorescent signals was made with particle analysis, by running a pre-recorded command line. This macro was written and optimized (S. Z.), according to our study protocol. As a result, we identified and saved the regions of interest (ROI) and projected them to our original images. Moreover, the mean grey values were measured (proportional to fluorescent intensities). These values were corrected to the background fluorescence by quantifying five independent, same-sized non-antibody-specific fluorescent areas in each image. The data were summarized in MS^®^ Excel, and the changes in mean fluorescence intensity (MFI) were reported relative to the control values. The importance of this method is based on the usage of clinical samples. Since the sample size is limited, however, the fairly large number of the quantified regions guarantees a solid data set. Additionally, the confocal microscopic approach provides specific measurements of the endothelial cell population. Nevertheless, in order to validate our evaluation protocol, we also applied the NOS3- and P-NOS3-specific immunocytochemistry (ICC) methods on isolated HUVEC suspensions, followed by FACS analysis.

### 2.8. Total NO_2_ and NO_3_ (tNOx) Measurement with Griess Reaction 

UC arteries and veins were freshly dissected at 4 °C. The vessels were snap-frozen and powdered under liquid nitrogen. The pulverized tissues were homogenized in an ice-cold 0.9% (*w*/*v*) salt solution and then centrifuged (4 °C, 17,000 g, 20 min). The supernatant was kept at −80 °C until the Griess assay.

To determine the tNOx content, the Schmidt and Kelm’s micro-assay protocol was followed with minor modifications [31]. Briefly, homogenates were diluted five-fold with an aqueous EDTA (Sigma-Aldrich, St. Louis, MO, USA: E5134) solution (100 mM), deproteinized with a short heat treatment at 75 °C and centrifuged (4 °C, 17,000 g, 20 min). The enzymatic reduction of the supernatants’ NO_3_ content to NO_2_ was carried out by 0.1 U/mL nitrate reductase (Sigma-Aldrich, St. Louis, MO, USA: N7265) in the presence of 0.03 mM NADPH (Apollo Scientific, Manchester, UK: BIB3014) and 5 µM FAD (Sigma-Aldrich, St. Louis, MO, USA: F6625) cofactors (37 °C for 15 min). To terminate the reaction, the cofactors were oxidized with 1 mM potassium ferricyanide (Fisher Scientific, Loughborough, UK: P/4880/53), and the samples were cooled down and kept at 4 °C. The diazotization was carried out in 1 mM sulphanilamide (Sigma-Aldrich, St. Louis, MO, USA: S9251) and 0.6 M hydrochloric acid (VWR, West Chester, PA, USA: 20252.290), and then centrifuged (4 °C, 3500 g, 15 min). The supernatant was taken for the azo-coupling step and treated with 1 mM *N*-(1-naphthyl)ethylenediamine (Sigma-Aldrich, St. Louis, MO, USA: 222488). The same protocol was used for creating the calibration curve, where 1 mM sodium nitrite (Sigma-Aldrich, St. Louis, MO, USA: 237213) was used in a serial dilution (100 µM–6.25 µM). The absorbance of the resulted stable azo dye was read at 540 nm, using a GENESYS 10S UV-Vis spectrophotometer (Thermo Fischer Scientific, Madison, WI, USA). The tNOx concentrations were calculated from the calibration curve, considering the dilution factor. Results were given relative to the sample’s initial protein concentration and expressed as nmol/mg protein [32].

### 2.9. Statistical Analysis and Graphic Representation

For statistical analyses and graphs, GraphPad Prism version 6.00 for Windows was used (GraphPad Software, La Jolla, CA, USA). All the statistics were carried out on the measured parameters, without normalization. The data sets of semi-quantified immunolabeling, the oxidized DHE levels and the tNOx concentrations were evaluated by an unpaired *t*-test followed by a Mann–Whitney test to compare ranks. For the viability assay, a grouped analysis with a two-way ANOVA, followed by a Holm–Sidak’s multiple comparisons test, was applied. For the qPCR results, we applied a one-way ANOVA, followed by a Tukey’s multiple comparisons test on our ΔCt values. Significant differences were accepted at * *p* ≤ 0.05, ** *p* ≤ 0.01, *** *p* ≤ 0.001 and **** *p* ≤ 0.0001.

## 3. Results

### 3.1. Morphological Changes in the Endothelial Layer of the UC Vessels Indicate Loss of Function and Cell Death

The endothelium of UC vessels is in direct contact with all the toxic/harmful components in the feto-placental circulation. Therefore, morphological and molecular changes in it may have direct effects on embryonic development. The functional endothelial layer in the control vessels can be characterized by close intercellular connections and intact cellular components (Figure 1a). Meanwhile, in the samples with smoking origin, many of the endothelial cells lose their intercellular contacts and exhibit a “dying” phenotype, most likely due to long-term adverse conditions. The common phenotypical features of these samples are their nuclear fragmentation, cytoplasmic shrinkage, detachment from the basement membrane and the formation of huge inter- and intracellular vacuoles in both vessels (Figure 1b,c). Veins are always affected to a greater extent, probably due to their primary exposition to toxic material unfiltered by the placenta (Figure 1c). The integrity of cord endothelial cells is important for the regulation of the vascular tone, thus providing a proper oxygen and nutrient supply to the developing fetus. The loss of functionality in the vein endothelium was followed by a viability assay. According to our results, only one-third of the isolated Sm vein endothelial cells were intact, and 65% were either in the early or the late apoptotic phase, while in the Ctrl samples, only 30% of the total cell population exhibited apoptotic properties (Figure 1d and Appendix A). In spite of the massive cell death, the UC vessels were still functional. The stress response capability of the cells was examined in an ex vivo experiment, where an isolated cord vein was subjected to heat and, in parallel, to a heat- and cadmium treatment. The expression of the hsp90 gene was followed by qPCR amplification, and the mRNA levels were increased about to the same extent (~two-fold) in both sample groups. In the combined stressor experiment, gene expression was induced by Cd^2+^ treatment in a concentration and sample-specific manner. In the case of 0.5 ng/μL Cd^2+^, there was a ~ seven-fold elevation in the control samples and a ~ three-fold increase in the samples with smoking origin. Cadmium ions at higher concentrations (2.5 ng/μL) were most likely toxic in both systems (Figure 1e). These data not only reinforced the reactivity of the cells in the UC vessels but can also serve as indirect evidence of a former Cd^2+^ exposure due to maternal smoking.

### 3.2. Endothelial Dysfunction of the UC Vessels Accompanied by Membrane and Protein Damage

One of the main reasons behind ED is the oxidative stress via ROS and reactive nitrogen species accumulation. To evaluate the level of oxidative stress, we followed the intracellular O_2_^•−^ level and the macromolecular damages. The common starting point of oxidative stress, the O_2_^•−^ formation, was detected with an in situ DHE assay. The specificity was tested prior to the application (Appendix A). In order to follow the protein and membrane damages, a specific immunolabeling for lipid peroxidation product, 4-hydroxy-2-nonenal (4-HNE), and the nitrosative stress marker, 3-nitrotyrosine (3-NT) modified proteins, were applied. DHE-derived fluorescence was markedly elevated with about the same extent in the Sm arteries and veins (~70–60%). As for the 4-HNE protein adducts formation: in the vein, the SmHNE level was more than twice of the values measured in the arteries (~270% versus ~125%). The nitrosylation of proteins remained nearly unchanged in the arteries, while in the veins it showed a three-fold increase (~300%), compared to the controls (Figure 2a,b and Appendix A).

### 3.3. NOS3-Dependent NO Production Is Altered under Severe Oxidative Stress Condition

The intact/functional endothelial layer of the UC vessels is pivotal for NO production by NOS3, along with the vasodilator mechanism and many other important biological functions. NOS3 expression and its phosphorylation status at SER-1177 were followed on double immunolabeled (anti-NOS3/P-NOS3) cryosections, originated from Ctrl and Sm UCs. In general, we found that either the NOS3 expression or its phosphorylation level decreased in the Sm vessels in comparison with the controls. In the arteries, it was the NOS3 protein level that dropped drastically (~55%), while in the vein there was no noteworthy difference in the expression level (~2%). The NOS3-related pSER1177 revealed a comparable degree of reductions (~37–45%) in the two types of vessels (Figure 3a,b and Appendix A). These findings suggest an impaired NO production in both Sm vessels; in the arteries, we found a down-regulated expression of NOS3, with an increased phosphorylation level, while in the veins the NOS3 protein level was nearly unchanged but the phosphorylation dropped significantly. To validate the IHC data set, ICC was performed on isolated HUVEC populations using anti-NOS3/anti-P-NOS3 labeling, which was quantified by FACS analysis. The purity of the isolated endothelial cell population was higher than 95%, based on the analyses of anti-vWF/anti-NOS3 double immunolabeling (Appendix A). 

In parallel with the evaluation of NOS3 and its phosphorylation level, ARG1 and NOS2 expressions were also followed, both being crucial influencers of NO production under oxidative stress condition. In the arteries we found a marked increase (~130%) in the SmARG1 level, while the SmNOS2 level decreased by ~40% (Figure 3c and Appendix A). In the case of the veins, both the ARG1 and NOS2 expressions exhibited a significant increase (~40% and ~50%, respectively) in comparison to the controls (Figure 3d and Appendix A).

### 3.4. Xanthine Oxidoreductase Potentially Contributes to NOS3-Independent NO Production in Case of Endothelial Dysfunction

NOS3 enzyme activity is fully dependent on O_2_. Consequently, under hypoxic condition an activation of compensatory mechanisms could function as a rescue mechanism for the NO signaling pathway. XOR has a capacity to reduce inorganic NO_3_ to NO_2_ under conditions of low oxygen tension. The concentration of total nitrogen oxide derivatives (tNOx) and the XOR content were measured in the cord vessels with both origins. An increased content of SmtNOx was detected in both vessels, compared to Ctrl; the elevation was ~65% in the artery and ~140% in the vein (Figure 4a,b and Appendix A). Similarly, XOR level was also increased by ~30% in the artery and ~580% in the vein (Figure 4a,b and Appendix A). Based on these results, it is most likely that one way to compensate NO deficiency due to ED is the conversion of inorganic NO_2_/NO_3_ to NO by XOR.

## 4. Discussion

Smoking is still one of the main risk factors for the development of cardiovascular complications. Smoking-induced anomalies are well studied mostly in adults and in UC-derived in vitro cell culture systems [3,33]. In adults, most of the developing pathological conditions can be traced to the decreased level of bioavailable NO, due to ED [2,34,35]. In fetal development, it was recently demonstrated that even a moderate maternal smoking decreases the viability of fetal mononuclear cord blood cells and induces damages in UC arteries [36]. Nevertheless, their molecular background is still unclear. Moreover, it is also important to emphasize that smoking during pregnancy not only affects fetal development, but might also induce long-term health consequences such as elevated blood pressure and cardiovascular diseases in the offspring [37,38]. Furthermore, animal studies conducted by Lawrence and colleagues presented evidence of reprogrammed in utero gene expression, induced by prenatal nicotine exposure, which ultimately led to an increased heart disease susceptibility in adulthood [39].

The functionality of the UC vessels, in particular their endothelial layer, is a determining factor for the proper oxygenation and development of the fetus. Complications affecting the cord tissues could have a direct effect on fetal development. Moreover, the detectable changes may serve as markers of the damage imposed on the developing fetus or may be associated with the development of other diseases in later life. Therefore, the primary goal of our study was a comprehensive structural and molecular analysis on the UC vessels impacted by toxic material originated by maternal cigarette smoke. Such complex surveys on the cord vessels are sporadic in the literature, though their study has diverse advantages over studying the placenta and the UC blood. The measured cord blood parameters most likely reflect the circulating toxic agent content and oxidative status of neonates at the time of birth. The harmful agents accumulating in the placenta, of embryonic and maternal origin, do not necessarily enter the embryonic/fetal circulation, and the alterations detected in the placenta do not necessarily reflect the actual effect on the fetus. In contrast, the UC is fully embryonic in origin, and the cord vessels can be considered as an elongation of the vascular system of the developing fetus. On the one hand, alterations in the UC vein clearly indicate the long-term appearance of harmful substances in the fetal circulation and might serve as fingerprints of the damages affecting the developing fetus. On the other hand, changes in the physiological state of the cord have a direct negative influence on the fetal development.

Based on the characteristic phenotypical parameters of our study populations, it is clear that maternal smoking impacts fetal development. Significantly lower head and chest circumference values (6–6.5%) were measured in cases of neonates born to smoking mothers. The birthweight-based distribution was even more characteristic: 55% of neonates in the control group had a birthweight above 3500 g, while only 13% of neonates with smoking origin reached this value [16].

Additionally, the structural and molecular data presented in this paper clearly indicate that continuous maternal smoking induces alterations not only in the vein endothelium, but also in the arteries, pointing out a general functional impairment. Furthermore, the results also indicate that the extents of the above-mentioned changes are in good correlation with the level of sustained exposition to harmful materials, in a vessel-specific manner.

Finally, the presented data set points out a correlation between the degrees of ROS-mediated macromolecular damages and the activation of alternative NO-producing pathways. These compensatory mechanisms are also induced in a vessel-specific manner. Because of the lack of innervation in the UC vessels, vasodilation is mainly regulated by the bioavailable NO, primarily produced by the activated NOS3. An intact plasma membrane has a pivotal role in the initial steps of NOS3 activation, and any damage to it leads to a compromised NO production [40]. The detected ultrastructural changes in the vessel’s endothelium indicate that the vein is more severely affected than the artery, most likely because it is primarily exposed to the toxic materials. The presented molecular data set on the investigated parameters collectively support the above conclusion. First, the level of bioavailable NO, produced by the activated NOS3 pathway, is most likely insufficient in the vein, due to the inefficient phosphorylation of NOS3 at the SER1177 position. This conclusion is reinforced by the fact that the alternative NO-producing pathways, such as NOS2 and XOR, are highly upregulated in the veins which exhibit a high extent of cellular and macromolecular damages. At first glance, NOS2 upregulation seems beneficial to the vein, providing an alternative source for bioavailable NO. However, the activity of NOS2 has other consequences, related to cytokine stimulation and ROS elevation [41]. Furthermore, elevated NOS2 activity is also linked to inflammatory processes and induces cell apoptosis through caspase-3 activation [42,43]. Our findings through viability assays on isolated vein endothelial cells and the damages visualized by electron microscopy support the conclusion that NOS2 activation is not necessarily beneficial to the cells. Indeed, the macromolecular damages in the vein endothelium indicate that the boosted production of NO by upregulated NOS2 leads, rather, to an increased nitrosative stress, instead of an increase in bioavailable NO. This nitrosative burden could evolve from the spontaneous reaction of NO and O_2_^•−^ to form peroxynitrite, a highly reactive cytotoxic oxidant, which could rapidly decompose to NO_2_ [44]. The peroxynitrite-mediated oxidation and/or nitration of biomolecules results in the elevated formation of 4-HNE in the vein and an increment in the 3-NT level, which correlated with the increase of NOS2 expression. This data set indicates that NOS2-derived NO is utilized for vasodilatation only to a limited extent, rather than providing the increased tNOx pool. We assume that the elevated tNOx pool in a hypoxic vein amplifies the tNOx-XOR-NO pathway, attempting to develop a compensatory mechanism for vascular impairment. Parallel to the NOS2 upregulation, the XOR level increased drastically in the vein, which supports the above conclusion. Going even further, it is not only a possible way; it might be the only one to convert the tNOx pool beneficial under this circumstance. In recent years, the active role of XOR in the offset of the hypoxic environment has received considerable attention [24]. However, we must acknowledge that in a highly oxidative condition every effort to increase the bioavailable NO level will reinforce this vicious circle, strengthening the oxidative/nitrosative stress. Considering the arteries, we have to remember that fetal circulation is peculiarly organized, with the arteries mainly carrying deoxygenated blood and fetal waste, partly mixed at multiple sites with umbilical-vein-derived oxygenated blood [7]. While cord veins are the primary targets for harmful substances unfiltered by the placenta, arteries are secondary targets and are mostly exposed in an indirect way. The state of the arteries probably reflects those effects, which have already impacted the fetus. In total, we found arteries in a less damaged condition than the veins, both from a structural and a molecular point of view. Despite the lower NOS3 level, it is more likely that the NOS3-mediated NO production is more or less satisfactory in the arteries. One supporting evidence is the highly increased phosphorylation status of the available NOS3 at the SER1177 position, paralleled by the lack of induction of the secondary NO-producing pathway. As we found it, NOS2 expression was even lower than in the matching control, and the XOR level was lower than the one detected in the veins. The upregulation of other pathways can also impair NOS3-NO production, mainly by competition for the common substrate L-arginine. One of the key players in this contest is the abundant cytoplasmic ARG1. Under non-stressful conditions, ARG1 plays a role in the maintenance of cell proliferation [45]. However, an excessive ARG1 expression, induced by several agents (e.g., ROS), leads to the reduced availability of L-arginine, resulting in NOS3 uncoupling and further ROS accumulation, which cause vascular impairment [46]. The upregulated level of ARG1 in the vein endothelium triggers the uncoupling of NOS3, resulting in a deficiency of the activation phosphorylation. Pernow and his co-workers presented evidence of the high-level expression and activation of ARG1 in the vascular endothelium of type-2 diabetes mellitus patients. Moreover, they provided convincing proof of an intimate crosstalk between RBCs and the vessel endothelium, and the determining role of RBCs in governing cardiac function through arginase-dependent regulation [47]. In a recent publication, we described maternal-smoking-induced morphological and functional abnormalities in fetal RBCs, originated from the same cord vessels used in the present work. In that study we pointed out that in fetal RBCs with smoking origin, the NOS3-NO pathway is unavailable as a rescue mechanism in case of vascular dysfunction. Furthermore, we proved that the maternal-smoking-induced ARG1 expression in Sm-RBCs may even augment the vascular dysfunction and serve as the etiology for cardiovascular diseases [16,47].

## 5. Conclusions

Our datasets on the artery and vein endothelium clearly demonstrate that maternal-smoking-induced alterations are vessel-specific and most likely a close reflection of the in vivo circumstances of prenatal development. Furthermore, we prove that in a highly hypoxic environment, with a low bioavailability of the NO level, an elevated NO_2_/NO_3_ pool unravels the recently described protective role of XOR. Our ex vivo experiments point out that Sm-veins still have the capacity to respond under various stress conditions. Furthermore, these findings clearly indicate a sort of adaptation to Cd^2+^, as a consequence of a long-term exposure to the toxic components of cigarettes. Our results contribute to a better understanding of the molecular changes that occur upon intrauterine toxic exposure, and in-depth knowledge of the affected pathways may determine novel therapeutic targets for clinical and applied research. 

## Figures and Tables

**Figure 1 antioxidants-10-00583-f001:**
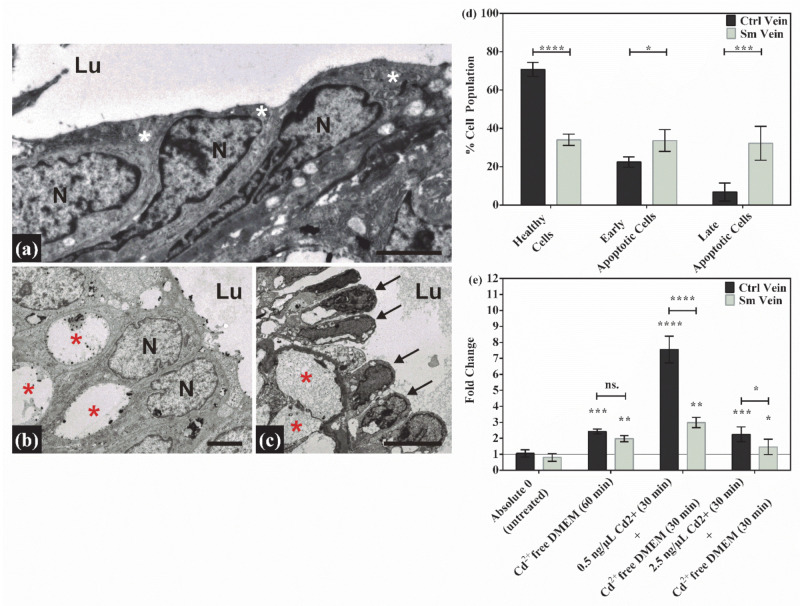
Morphological changes, viability and reactivity in Ctrl and Sm umbilical cord (UC) vessels. On the representative electron micrographs focusing on the endothelial layer, significant changes can be observed compared to the Ctrl (**a**) endothelial cells (white asterisks) with the Sm artery (**b**) and Sm vein (**c**) endothelial cells (black arrows). In addition to losing their intercellular contacts, cells also show damaged/dying phenotypic features in the cytoplasm and nuclei (*N*). The most remarkable is the nuclear fragmentation and the occupancy of intra- and intercellular vacuoles (red asterisks) at the expense of the cytoplasm. Scale bars: 2 µm (**a**); 2 µm (**b**); 5 µm (**c**). (Lu = vessel lumen). (**d**) Graphical presentation of the viability assay data set on isolated Ctrl- and Sm-derived human umbilical vein endothelial cell (HUVEC) populations. Flow-cytometer-quantified data are ordered on the interleaved bar graph, where the percentage distribution of healthy, early and late apoptotic cells are expressed within the whole isolated Ctrl and Sm populations, and shown as mean ± SD. Total cell numbers were the following: Ctrl (*n* = 237500 ); Sm (*n* = 129600 ), from three and two individual samples, respectively. Statistics: grouped analysis with two-way ANOVA, followed by Holm–Sidak’s multiple comparisons test. * *p* ≤ 0.05, *** *p* ≤ 0.001 and **** *p* ≤ 0.0001. (**e**) Bar graph showing the changes in HSP90 mRNA expression, upon heat and combined (heat + Cd^2+^) treatment in Ctrl- and Sm-derived UC veins. Ctrl (*n* = 2) and Sm (*n* = 2) samples tested in triplicates. Statistics: grouped analysis with two-way ANOVA, followed by Holm–Sidak’s multiple comparisons test. ns. = non – significant, * *p* ≤ 0.05, and ** *p* ≤ 0.01.

**Figure 2 antioxidants-10-00583-f002:**
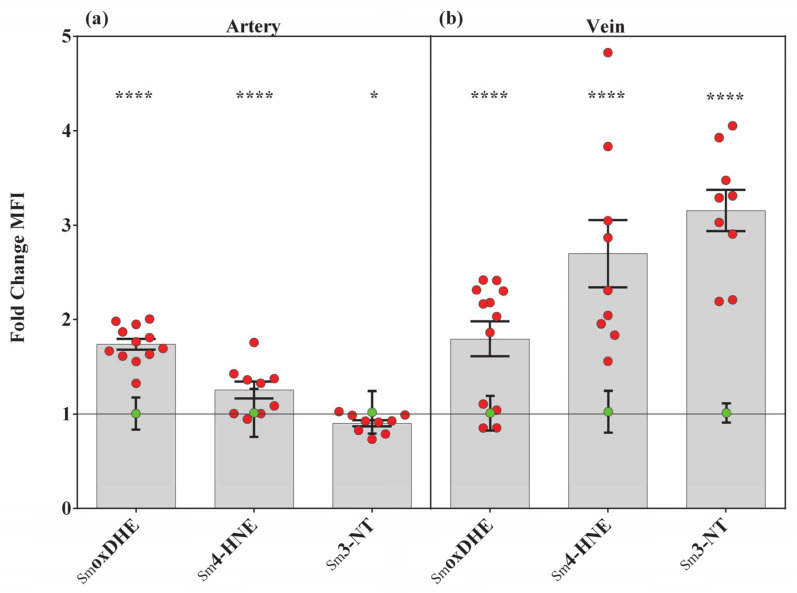
Changes in oxidative stress markers among Sm-derived UC vessels. Bar graphs showing the semiquantified results of 4-HNE, 3-NT specific immunolabeling and the intercalated redox indicator dihydroethidium staining (DHE) levels in Sm arteries (**a**) and in Sm veins (**b**). Red dots symbolize the overall mean fluorescence intensity (MFI) values from individual Sm samples, while green dots refer to summarized data from the Ctrl group, applied as the baseline of comparison (mean ± SEM). Sample numbers: CtrloxDHE (*n* = 304 artery and 929 vein regions of interest (ROI) from 12 individual samples); SmoxDHE (*n* = 961 artery and 1229 vein ROI from 12 individual samples), Ctrl4-HNE (*n* = 233 artery and 589 vein ROI from 11 individual samples); Sm4-HNE (*n* = 232 artery and 448 vein ROI from 9 individual samples); Ctrl3-NT (*n* = 1033 artery and 44 vein ROI from 11 individual samples); Sm3-NT (*n* = 277 artery and 39 vein ROI from 9 individual samples). Statistics: unpaired *t*-test followed by Mann–Whitney test to compare ranks * *p* ≤ 0.05, **** *p* ≤ 0.0001.

**Figure 3 antioxidants-10-00583-f003:**
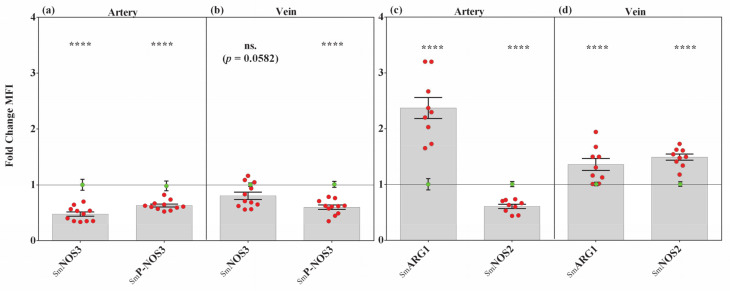
Changes of NO synthesis determining factors in Sm-derived UC vessels. Bar graphs showing the semiquantified results of NOS3-, P-NOS3-, ARG1- and NOS2-specific immunolabeling in Sm arteries (**a**,**c**) and in Sm veins (**b**,**d**). Red dots symbolize the overall MFI values from individual Sm samples, while green dots refer to the summarized data from the Ctrl group, applied as the baseline of comparison (mean ± SEM). Sample numbers: CtrlNOS3 (*n* = 95 artery and 112 vein ROI from 12 individual samples); SmNOS3 (*n* = 53 artery and 102 vein ROI from 11 individual samples), CtrlARG1 (*n* = 77 artery and 781 vein ROI from 10 individual samples); SmARG1 (*n* = 64 artery and 732 vein ROI from 9 individual samples); CtrlNOS2 (*n* = 211 artery and 130 vein ROI from 11 individual samples); SmNOS2 (*n* = 100 artery and 243 vein ROI from 9 individual samples). Statistics: unpaired *t*-test followed by Mann–Whitney test to compare ranks ns. = non-significant, **** *p* ≤ 0.0001.

**Figure 4 antioxidants-10-00583-f004:**
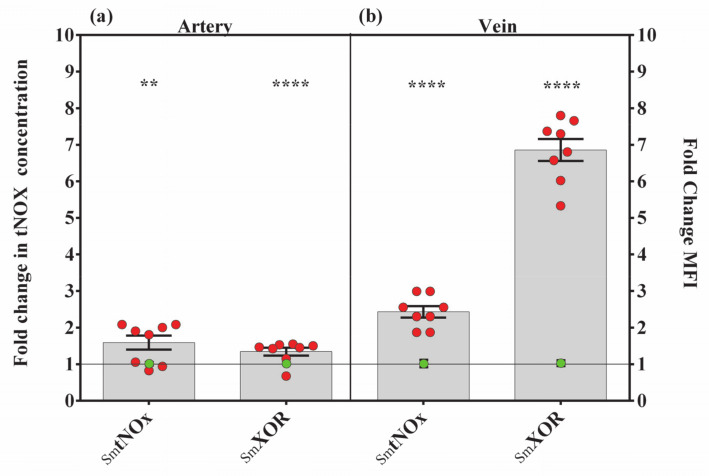
Changes in NOS-independent NO production in Sm-derived UC vessels. Bar graphs show the semiquantified results of xanthine oxidoreductase (XOR)-specific immunolabeling and the tNOx concentrations determined by a Griess microassay in Sm arteries (**a**) and in Sm veins (**b**). Red dots symbolize the overall MFI or tNOx concentration values from individual Sm samples, while green dots refer to the summarized data from the Ctrl group, applied as the baseline of comparison (mean ± SEM). Sample numbers: CtrlXOR (*n* = 44 artery and 83 vein ROI from 8 individual samples); SmXOR (*n* = 33 artery and 19 vein ROI from 8 individual samples). In case of CtrltNOx (*n* = 8 individual samples) and SmtNOx (*n* = 8 individual samples) measurements carried out in triplicates. Statistics: unpaired *t*-test followed by Mann–Whitney test to compare ranks, ** *p* ≤ 0.01, **** *p* ≤ 0.0001.

## Data Availability

Available upon request.

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
