# Peer review of "Sustained Maternal Smoking Triggers Endothelial-Mediated Oxidative Stress in the Umbilical Cord Vessels, Resulting in Vascular Dysfunction"

_antioxidants, 2021, doi:10.3390/antiox10040583_

Round 1

Reviewer 1 Report

Zahorán et al. have performed an interesting study about how a sustained maternal smoking triggers endothelial-mediated oxidative stress in the umbilical cord vessels. 

However, I have a pair of considerations, that I would like authors answer me

  1. Have authors thought to analyze this parameters with cotinine 
    cord blood determination? Cotinine has a longer half-life than nicotine, and cotinine concentrations in serum, urine, hair, and saliva are commonly used as biomarkers of recent tobacco exposure in epidemiological studies (Drug Alcohol Depend. 2010, 107 (2–3): 250-252; Nicotine Tob Res. 2003, 5 (3): 349-355. ). Lack of a placental barrier for cotinine (and probably nicotine) can partially explain smoking-related perinatal disorders ( Eur J Obstet Gynecol Reprod Biol. 2012, 165 (2): 205-209; Drug Alcohol Depend. 2010, 107 (2–3): 250-252)
  2.  Could authors show a representative histograms from flow cytometry analysis in umbilical cord mononuclear blood cells from non-smokers and smokers?
  3. MFI is twice defined

Reviewer 2 Report

The manuscript by Zahoran and colleagues describes effects of smoking on nitric oxide signaling pathways in the umbilical vein and artery. Segments of umbilical vein were isolated from heavy smokers (> 10 cigarettes/day) and non-smokers. Morphological changes, including the expression of apoptotic markers, were noted in umbilical veins of smoking mothers, indicative of endothelial cell damage. These changes were reduced in umbilical arteries. Formation of superoxide anion was elevated in both arteries and veins of smokers. Additionally, a two-fold increase in lipid peroxide formation was noted in veins compared to arteries. Protein nitrosylation was increased three-fold in veins of smokers compared to non-smokers. NOS3 protein levels were unchanged veins of smokers compared to controls, but the level of phosphorylation was significantly reduced, indicating a reduction in NO formation. Further, arginase and NOS2 levels were increased. tNOx was elevated in both arteries and veins of smokers compared to non-smokers. The authors observed an increase in XOR in veins and suggest that this pathway can increase NO bioavailability.

Reviewer 3 Report

The original article by Zahorán et al. is nicely written and aims to add new knowledge about the effect of sustained maternal smoking in triggering the endothelial-mediated oxidative stress in the umbilical cord vessels, resulting in vascular dysfunction.

A large number of interesting experiments have been performed and the results seem in line with the purpose of the article.

However, although it is glimpsed in the title, at the end of the discussion the purpose of the article becomes evanescent or it is not clearly stated, therefore, without the title, the aim of the article is unknown.

Also, in my opinion, the authors fail in bringing out the originality of their results (if any) compared to other research on the same topic.

I believe that these aspects should be addressed before publishing the article.

Round 2

Reviewer 1 Report

I don't have more questions to be answered. Authors have solved answered all my doubts